# AI-Assisted Composite Etch Model for MPT [note 1]

**DOI:** 10.3390/mi16121410

**Published:** 2025-12-15

**Authors:** Yanbin Gong, Fengsheng Zhao, Devin Sima, Wenzhang Li, Yingxiong Guo, Cheming Hu, Shengrui Zhang

**Affiliations:** 1Dongfang Jingyuan Electron Co., Ltd., Beijing 100176, China; yanbin.gong@dfjy-jx.com (Y.G.); fengsheng.zhao@dfjy-jx.com (F.Z.); 2Fujian Jinhua Integrated Circuit Co., Ltd., Quanzhou 362200, China; devin.sima@jhicc.com (D.S.); wenzhang.li@jhicc.com (W.L.); yingxiong.guo@jhicc.com (Y.G.); vincent.hu@jhicc.com (C.H.)

**Keywords:** SEM contour, etch model, multiple patterning, LELE, SARP, ERC, hotspot detection, auto retargeting

## Abstract

For advanced semiconductor nodes, the demand for high-precision patterning of complex foundry circuits drives the widespread use of Lithography-Etch-Lithography-Etch (LELE)—a key Multiple Patterning Technology (MPT)—in Deep Ultraviolet (DUV) processes. However, the interaction between LELE’s two Lithography-Etch (LE) cycles makes it very challenging to build a model for etching contour simulation and hotspot detection. This study presents an Artificial Intelligence (AI)-assisted composite etch model to capture inter-LE interactions, which directly outputs the final post-LELE etch contour, enabling Etch Rule Check (ERC)-based simulation detection of After Etch Inspection (AEI) hotspots. In addition, the etch model proposed in this study can also predict the etch bias of different types of pattern (especially complex two-dimensional (2D) patterns), thereby enabling auto retargeting for After Develop Inspection (ADI) target generation. In the future, the framework of this composite model can be adapted to the Self-Aligned Reverse Patterning (SARP) + Cut process to address more complex MPT challenges.

## 1. Introduction

Among the multiple patterning techniques, Lithography-Etch-Lithography-Etch (LELE) stands as an essential practical method for overcoming the resolution limitations of 193 nm lithography [1]. It enables the fabrication of high-density patterns with sub-40 nm half-pitch by decomposing the original design layout into two or more complementary mask sets. These masks are then used in multiple sequential Litho-Etch cycles, effectively surpassing the resolution limits of single-exposure Deep Ultraviolet (DUV) lithography without requiring Extreme Ultraviolet (EUV)-based processes.

The core principle of LELE involves decomposing complex patterns and realizing them through multiple cycles of exposure and etch, though this comes at the cost of demanding overlay control requirements and increased process complexity [2,3,4]. However, the resist and etch models commonly used in the industry are only capable of simulating the patterns manifested on the wafer after a single exposure and etch process. Up to now, research on a composite model that accurately describes the complete dual Lithography-Etch (LE) process has not yet been reported. The inter-LE-cycle coupling makes it extremely difficult to accurately simulate the complete LELE process and predict the final wafer contour.

To address this limitation, we first turned to Scanning Electron Microscope (SEM) contour analysis—a metrology method uniquely suited to characterize pattern variability beyond simple Critical Dimension (CD) measurements [5]. Unlike CD gauges, which only quantify linear dimensions, SEM contours capture the full two-dimensional (2D) shape of patterns. In this work, we extracted high-fidelity SEM contours at key LELE steps: LE1 After Develop Inspection (ADI), LE1 After Etch Inspection (AEI), LE2 ADI, and LE2 AEI. These contours serve as the foundational data for model calibration and validation, as they capture the precise pattern evolution across both LE cycles.

The key innovation of this work lies in integrating these SEM contour data with Artificial Intelligence (AI). Traditional modeling frameworks, which rely solely on physical terms, cannot fully resolve the complex, non-linear interactions between LE1 and LE2. By embedding small-scale neural networks (dubbed AI-Assisted Flexible Terms, AAFTs) into the modeling workflow, we developed an AI-assisted composite etch model that explicitly captures the coupling between the two LE cycles. This model not only overcomes the limitations of single-cycle models but also achieves what was previously unfeasible: directly outputting the final post-LELE etch contour.

In summary, this work establishes a new paradigm for LELE process simulation. By combining SEM contour metrology (for high-quality input data) and AATFs (for resolving inter-LE interactions and enhancing model fitting ability), we bridge the long-standing gap in full LELE simulation. The resulting model enables not only the accurate prediction of the final etched contour but also the early detection of AEI hotspots via Etch Rule Check (ERC). Importantly, this workflow can be extended to more complex Multiple Patterning Technology (MPT) scenarios, such as Self-Aligned Reverse Patterning (SARP) combined with Cut processes.

## 2. Materials and Methods

### 2.1. Process Flow

This case study presents the metal layer patterning flow using LELE process of ArF-immersion, and Numerical Aperture (NA) is 1.35, mask type is opaque MoSi on glass (OMOG), with customer illumination, sector (X-Y) TE-polarized, and negative-tone develop (NTD) resist process [6].

Figure 1 illustrates the flow of the LELE process, which involves two sequential Litho-Etch cycles followed by a final etch step. LE1 is identical to a single Litho-Etch process; The final etch step is essentially a global etch process that precisely transfers the combined pattern of LE1 and LE2 to the metal layer, with negligible (effectively 0 nm) CD variation. The challenge lies not in LE1 and final etch, but in the coupled effects between LE1 and LE2. Once the LE1 pattern has been transferred onto the hard mask, it forms an uneven incoming layer that influences both the exposure and etch process of LE2. Under these conditions, LE2 behaves differently compared to a single Litho-Etch process, resulting in a discrepancy between the LE2 wafer CD and the design CD. This deviation is referred to as CD variation. Additionally, during the strip step for photoresist removal in LE2, the process may interact with the LE1 pattern on the hard mask, leading to LE1 CD loss. Both LE2 CD variation and LE1 CD loss are undesirable from a design perspective, which probably lead to CD out of spec. In severe cases, this may lead to defects such as pinch or bridge. Covering these potential defects using traditional rule-based bias table methods is highly challenging. A more comprehensive approach involves using an LELE model to simulate the final LELE contour, followed by performing ERCs.

### 2.2. Data Collection

Hitachi CD-SEM CG6300 (Hitachi High-Tech Corporation, Tokyo, Japan) was utilized for image generation and CD measurements, coupled with the DG-A software VR24.00 for contour extraction [6]. For each measurement point, five sets of SEM images are captured. First, the contour is extracted from each SEM image individually, and then these five contours are averaged to obtain the final average contour for calibration and validation. Contours of test patterns at identical locations on both LE1 and LE2 were extracted, and point gauges were generated along the normal direction of the ADI/AEI contours for corresponding model calibration and validation. Among these, about 11,000 gauges were used for model calibration, and 4000 gauges were reserved for validation.

### 2.3. Model Building

Existing methodologies for Optical Proximity Correction (OPC) and etch modeling are primarily semi-empirical approaches based on terms (which can be understood as filters in image processing). In modeling practice, the essence lies in selecting multiple terms with certain physical significance, combining them, and processing the input image to obtain the output images of each term. These output images are then superimposed to produce the final output image. Subsequently, a threshold is applied to the final output image for binarization, thereby enabling the derivation of the ADI or AEI contour. The selection of terms and their combination are determined by fitting to actual measured wafer data. By specifying the terms and their weighting coefficients, a specific ADI or AEI contour can be generated, from which CD can be measured. By comparing the model CD with the measured wafer CD, an evaluation function for the fitting process is established. The modeling process involves continuously adjusting the term coefficients and combination methods to ensure that the computational results align as closely as possible with the measured wafer data. The entire procedure is highly analogous in principle to polynomial fitting.

Due to the complex physical and chemical effects involved in the etch process, achieving high model accuracy solely through traditional terms is challenging [7,8,9,10,11]. With the rapid development of AI, there have been recent reports on utilizing AI algorithms to improve the accuracy of etch models [12,13]. Notably, the PanGen, a computational lithography product of DJEL (Dongfang Jingyuan Electron Co., Ltd., Beijing, China), is entirely developed based on a hybrid Graphics Processing Unit (GPU) + Central Processing Unit (CPU) supercomputing architecture and Compute Unified Device Architecture (CUDA). It operates in the same computational environment as AI frameworks, enabling the seamless integration of AI programs into the core functions of computational lithography. This inherent design allows PanGen to more effectively capitalize on the rapid advancements in AI technology, combining powerful AI algorithms with domain-specific expertise in computational lithography to deliver more efficient and intelligent solutions.

To enhance etch model accuracy, we have innovatively introduced small-scale neural networks as supplementary terms into the existing modeling framework, referred to as AAFT (Figure 2). Powered by AI engines, AAFTs significantly enhance the model fitting capability, leading to significant improvements in accuracy, particularly for 2D patterns. To mitigate overfitting risks, (1) AAFTs are combined with explicitly physically traditional terms during model calibration and (2) massive amounts of SEM contour datasets. This dual approach effectively reduces model overfitting, ensuring robust generalization capability alongside enhanced fitting performance.

The composite model for the LELE process is composed of four sub-models, including the LE1 ADI/AEI model and the LE2 ADI/AEI model. For the ADI modeling of LE1 and LE2, the most widely used and mature photoresist modeling process can be followed. The LE1 AEI model, which is a single-etch model, can be effectively constructed using the aforementioned AI-assisted SEM contour-based modeling flow.

The composite nature of the model is reflected in the mutual coupling of inputs and outputs among the four sub-models. Specifically, the input to the LE1 ADI model is the LE1 aerial image. After being processed through relevant terms, it outputs the LE1 resist contour, which then serves as an input for calibrating the LE1 AEI model after being transform into an image internally. Subsequent term processing yields the M01 etch contour, which can further be used as part of the input for the LE2 AEI model.

The main challenge in building the LELE composite model lies in the L2 AEI model because of the interaction between LE1 and LE2. This study innovatively integrates the LE1 etch contour and the LE2 resist contour to generate a comb contour as one input channel. Meanwhile, the LE1 etch contour and the LE2 resist contour are also used as two additional independent input channels, resulting in a total of three input channels. These three channels are processed through their respective terms. Using the LE2 wafer AEI contour, which is also the LELE final contour, as the training label, the solver under the AI framework is employed for iterative optimization to ultimately obtain a model with the minimal possible root mean square error (RMS).

## 3. Results and Discussion

A direct comparison between the model’s calibration and validation results provides a straightforward indication of potential overfitting. The error of the composite model can be represented by the RMS metric, as detailed in Table 1, which summarizes the calibration and validation results of both the ADI and AEI models for layer 1 and layer 2. As shown in Table 1, the RMS for both calibration and validation are comparable across all four models, indicating a minimal risk of overfitting.

Figure 3, Figure 4, Figure 5 and Figure 6 present a comparative analysis of the contours extracted from ADI and AEI images at identical locations on the wafer alongside the corresponding simulated contours generated by the models, along with their respective model error distributions. It can be observed that all four models exhibit reasonable error ranges and distributions. Specifically, approximately 82% of the ADI model errors for both LE1 and LE2 fall within ±2 nm, while about 53% are within ±1 nm. Similarly, for the AEI models, nearly 80% of the errors for LE1 and LE2 are within ±2 nm, and approximately 50% are within ±1 nm. Furthermore, the point gauge’s errors exhibiting >2 nm are predominantly located at pattern corners or line ends. Notably, for the LE2 AEI model, the RMS without the AAFT is 2.26 nm, which decreases to 1.80 nm with the AAFT included—a reduction of approximately 20%. This clearly demonstrates the enhanced data fitting capability contributed by the AAFT. Based on the above analysis, the composite model developed for the LELE process demonstrates excellent agreement between simulated contours and experimentally extracted wafer contours, meeting all required accuracy specifications without exhibiting signs of overfitting.

Applying this composite model to simulate the contour, the final LELE contour appears smooth without any abnormal deformations such as waviness, voids, or protrusions. The simulated contour matches well with the wafer LELE contour, with the well-matched 2D patterns being particularly noteworthy—a result attributable to the strong fitting capability of the AAFT. The simulated contour demonstrates good consistency, with uniformity of less than 1 nm for identical patterns. It also shows negligible grid dependency, with an average grid dependency of less than 1 nm across all gauges. The simulation runtime is acceptable: for a layout area of 300 × 300 μm^2^, the LE1 ADI model runs in 244 s, the LE1 AEI model in 383 s, and the LE2 AEI model in 445 s. The runtime for the LE2 simulation is 1.82 times that of the LE1 ADI model, which remains within an acceptable range and is capable of supporting full-chip simulation.

## 4. Conclusions

This paper has presented a comprehensive AI-assisted and contour-based methodology for modeling the complex Litho-Etch-Litho-Etch (LELE) dual-patterning process. The results demonstrate that the calibrated models for all four stages (LE1 ADI, LE1 AEI, LE2 ADI, LE2 AEI, or final LELE) show excellent agreement between the simulated and extracted wafer contours. The RMS errors for calibration and validation are comparable across all models, indicating robust performance without overfitting. Furthermore, error analysis confirms that the majority of errors are within acceptable limits (±2 nm), with larger discrepancies predominantly occurring at challenging pattern features like corners and line ends. This composite model produces simulated contours with good consistency, negligible grid dependency, and an acceptable runtime.

In conclusion, the integrated model proves to be highly reliable for key computational lithography tasks: retargeting the design layout to ADI targets by compensating for etch bias and performing mask correction based on the ADI targets. This work provides a viable and accurate strategy for OPC model calibration in MPT, which is critical for ensuring high yield at advanced technology nodes.

## 5. Outlook and Future Work

For future work, on one hand, conceptually, the current LELE modeling flow can be reconstructed into a more elegant structure with three sub-etch models arranged in two levels—this breaks the existing LE2 etch model into two separate steps, as illustrated in Figure 7. Leveraging the same methodology, this composite modeling approach can be extended to more complex patterning schemes, such as the SARP + Cut process (shown in Figure 8), which involves five sub-models organized into four levels.

On the other hand, compared to traditional bias tables, the accurate etch model proposed in this work paves the way for developing an auto retargeting (ART) solution with higher accuracy and reduced labor costs. Figure 9 presents the ART flow. The starting point for ART is the set of segments generated by the original bias table-based retargeting process. Control points are placed on these segments to quantify the discrepancy between the resist model contour and the etch model contour. Fundamentally, this approach mirrors the method used in OPC, where control points are placed on segments to measure the edge placement error (EPE) between the resist model contour and either the ADI target or the ideal ADI contour.

Within the ART workflow, the ideal ADI contour is first fed as input into the etch model, which then generates the corresponding etch contour. The difference between these two contours (ideal ADI vs. generated etch contour) is calculated to quantify the etch bias. This bias value is subsequently used to update the initial ADI target, originally derived from traditional bias tables. Following one iteration, a revised ADI target is generated, which in turn yields an updated ideal contour. The iteration process is repeated until the newly computed etch bias converges closely with the pre-existing bias values, which can provide a much more accurate ADI target compared to the original one obtained from the bias table. Notably, this multi-iteration ADI target correction process—driven by the etch model—is analogous to mask optimization in OPC, which relies on a resist model. However, it represents a distinctly new scenario: in this workflow, the correction focuses on the ADI target itself, rather than the mask.

To conclude, the etch model’s ability to extend to complex MPTs (such as self-aligned patterning) and the ART flow together enhance the practical value of the proposed framework, thereby providing a reliable, scalable foundation for computational lithography in advanced semiconductor nodes, where high-precision patterning is of paramount importance.

## Figures and Tables

**Figure 1 micromachines-16-01410-f001:**
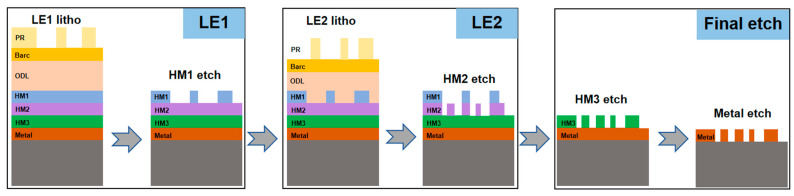
Litho-Etch-Litho-Etch process.

**Figure 2 micromachines-16-01410-f002:**
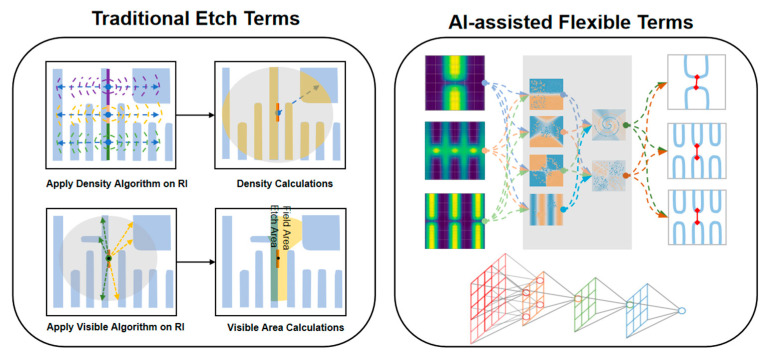
Traditional etch terms and AI-assisted flexible terms.

**Figure 3 micromachines-16-01410-f003:**
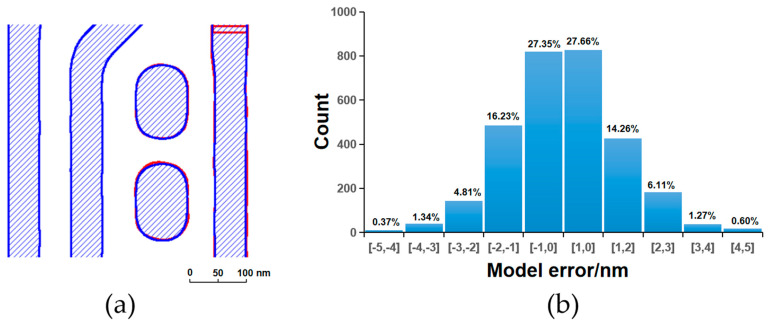
LE1 ADI, (**a**) contour simulation (blue filling) with SEM extraction (red contour); (**b**) model error distribution.

**Figure 4 micromachines-16-01410-f004:**
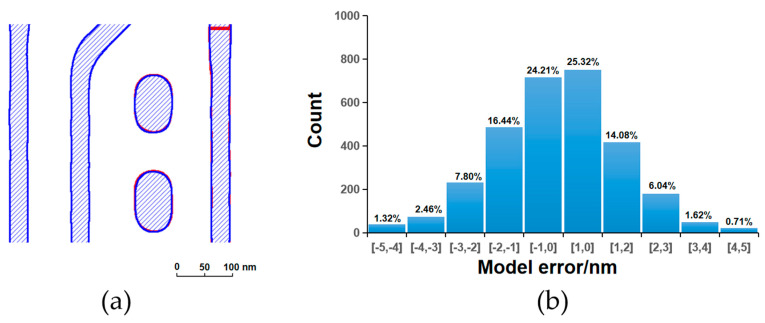
LE1 AEI, (**a**) contour simulation (blue filling) with SEM extraction (red contour); (**b**) model error distribution.

**Figure 5 micromachines-16-01410-f005:**
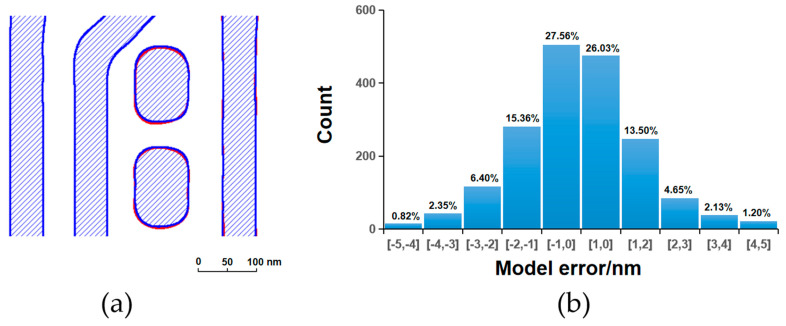
LE2 ADI, (**a**) contour simulation (blue filling) with SEM extraction (red contour); (**b**) model error distribution.

**Figure 6 micromachines-16-01410-f006:**
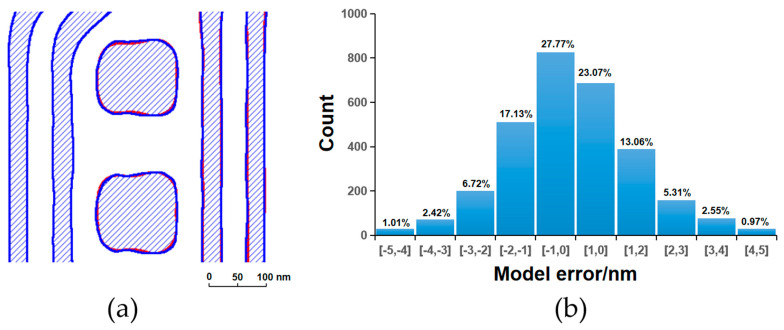
LE2/LELE AEI, (**a**) contour simulation (blue filling) with SEM extraction (red contour); (**b**) model error distribution.

**Figure 7 micromachines-16-01410-f007:**
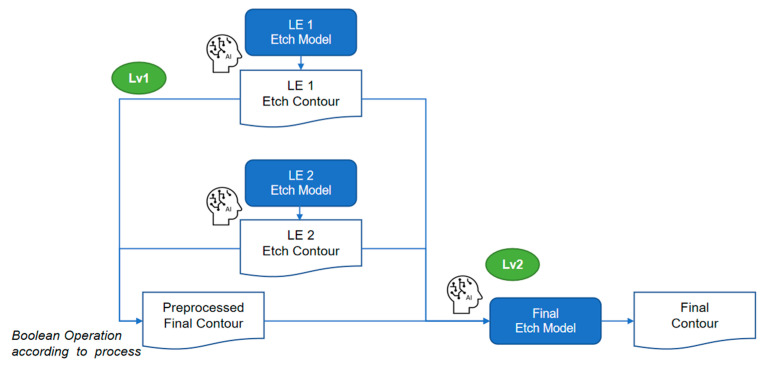
The extended composite model flow of LELE process.

**Figure 8 micromachines-16-01410-f008:**
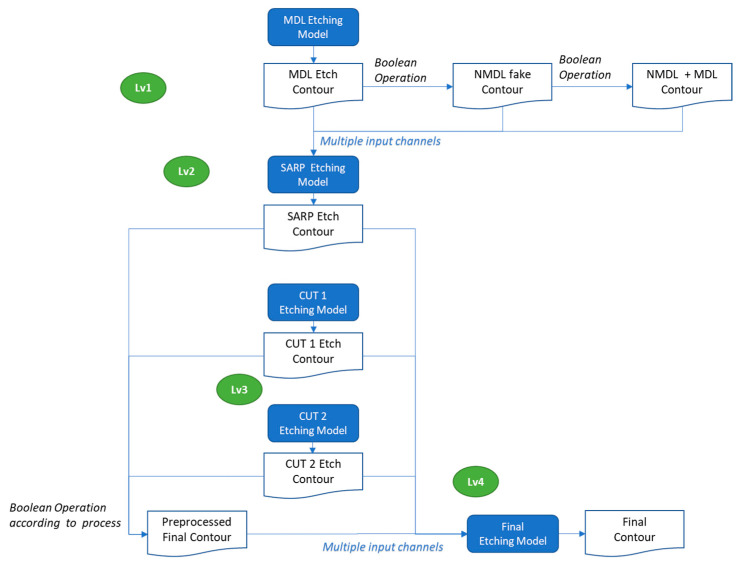
The extended composite model flow of SARP + Cut process.

**Figure 9 micromachines-16-01410-f009:**
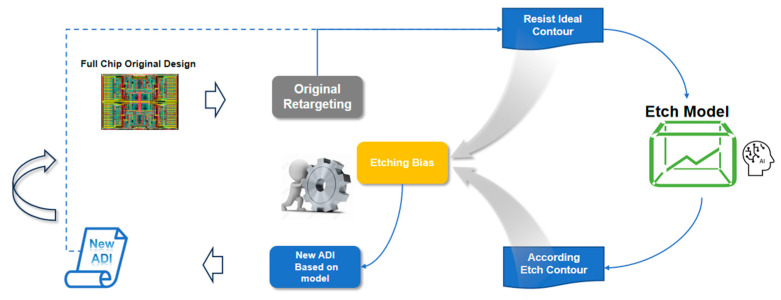
Auto retargeting flow illustration.

**Table 1 micromachines-16-01410-t001:** LELE combination composite model overall RMS.

Model	Calibration RMS/nm	Validation RMS/nm
LE1 ADI	1.42	1.50
LE1 AEI	1.70	1.82
LE2 ADI	1.62	1.71
LE2 AEI (Final LELE)	1.80	1.96

## Data Availability

The original contributions presented in this study are included in the article. Further inquiries can be directed to the corresponding author.

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
