# Peer review of "AI-Assisted Composite Etch Model for MPT [Author-notes fn1-micromachines-16-01410]"

_micromachines, 2025, doi:10.3390/mi16121410_

Round 1
Reviewer 1 Report
Comments and Suggestions for Authors
In this manuscript, the authors reported the AI-assisted composite etch model to capture insights of multiple patterning technology in DUVprocesses. This topic is very interesting and useful in IC fabrication. The calibrated models demonstrate excellent agreement between simulated and extracted wafer contours. However, the authors need to explain the modeling methods in the manuscript to help readers understand the process. Otherwise, this manuscript like an experimental report not a scientific paper. Furthermore, as the authers mentioned in the manuscript this composite model for LELE process, manuscript submitted for IWAPS. The authors need to check the data or figure is the same or not. The unit of X axis of the model error distribution should be provided in the figure.
Author Response
Comment1: [However, the authors need to explain the modeling methods in the manuscript to help readers understand the process. ]
Response1: [Here are the basic modeling methods for the ADI and AEI models, which have been incorporated into the model building section of the paper.
'Existing methodologies for Optical Proximity Correction (OPC) and etch modeling are primarily semi-empirical approaches based on terms (which can be understood as filters in image processing). In modeling practice, the essence lies in selecting multiple terms with certain physical significance, combining them, and processing the input image to obtain the output images of each term. These output images are then superimposed to produce the final output image. Subsequently, a threshold is applied to the final output image for binarization, thereby enabling the derivation of the ADI or AEI contour. The selection of terms and their combination are determined by fitting to actual measured wafer data. By specifying the terms and their weighting coefficients, a specific ADI or AEI contour can be generated, from which CD can be measured. By comparing the model CD with the measured wafer CD, an evaluation function for the fitting process is established. The modeling process involves continuously adjusting the term coefficients and combination methods to ensure that the computational results align as closely as possible with the measured wafer data. The entire procedure is highly analogous in principle to polynomial fitting.'
]
Comment2: [Furthermore, as the authors mentioned in the manuscript this composite model for LELE process, manuscript submitted for IWAPS. The authors need to check the data or figure is the same or not. ]
Response2: [The process flow and modeling flow are analogous; however, they are verified using different designs, and therefore do not share the same dataset. All figures and data presented in the paper are distinct from one another.]
Comment3: [The unit of X axis of the model error distribution should be provided in the figure.]
Response3: [The unit for the X-axis is nm and has been added, as shown in Figure 3-6.]

Reviewer 2 Report
Comments and Suggestions for Authors
This manuscript presents an artificial-intelligence-aided framework for simulating the Litho–Etch–Litho–Etch (LELE) process in multiple patterning technology. The authors integrate SEM contour-based metrology with AI-assisted flexible terms (AAFT) to construct a composite etch model capable of predicting final wafer contours and enabling etch-rule checks and auto-retargeting. The concept is technically interesting and aligns with current trends in data-driven computational lithography. However, the manuscript lacks actual experimental evidence such as SEM images or measurement data that would demonstrate how the model relates to real wafer results. In addition, several sections require clarification and improved methodological transparency to meet publication standards.
- Include representative SEM images or contour-extraction examples to illustrate the dataset used for calibration and validation.
- Provide a clearer explanation of how the AI-assisted flexible terms are implemented, including network structure, input features, training size, and loss function.
- Add quantitative benchmarking against a conventional single-cycle or rule-based etch model to demonstrate the improvement achieved.
- Provide basic information about the process stack and etch system so that readers can interpret the physical relevance of the model.
- Improve the clarity of Figures 3–6 by adding scale bars, higher resolution, and distinct colour coding to differentiate simulated and extracted contours.
- Expand the discussion on model applicability, limitations, and potential transferability to other pattern types or layers.
- Language and formatting corrections for grammar, article use, and spacing to improve readability.
Author Response
Comment 1: [Include representative SEM images or contour-extraction examples to illustrate the dataset used for calibration and validation. ]
Response 1: [As shown in Figures 3(a) to 6(a), the red contours represent the wafer contour extracted based on SEM, while the blue contours correspond to the model-simulated contour.]
Comment 2: [Provide a clearer explanation of how the AI-assisted flexible terms are implemented, including network structure, input features, training size, and loss function.]
Response 2: [The aforementioned comment involve proprietary information pertaining to core modeling parameters and cannot be disclosed. We trust you can appreciate our position. Thank you for your understanding.]
Comment 3: [Add quantitative benchmarking against a conventional single-cycle or rule-based etch model to demonstrate the improvement achieved.]
Response 3: [For the LE2 AEI model, the addition of the AI term yields a reduction in RMS from 2.26 to 1.80, corresponding to an approximate 20% decrease.]
Comment 4: [Provide basic information about the process stack and etch system so that readers can interpret the physical relevance of the model.]
Response 4: [As shown in Figure 1, the film stack and the LELE (Litho-Etch-Litho-Etch) process flow are illustrated.]
Comment 5: [Improve the clarity of Figures 3–6 by adding scale bars, higher resolution, and distinct colour coding to differentiate simulated and extracted contours.]
Response 5: [As shown in Figure 3-6, scale bars have been added and the resolution has been enhanced. To facilitate distinction, red and blue are used to represent the wafer contour and simulated contour, respectively.]
Comment 6: [Expand the discussion on model applicability, limitations, and potential transferability to other pattern types or layers.]
Response 6: [The content regarding model application is as follows and has been incorporated into the results and discussion section of the paper. The potential transferability to other pattern types or layers: MPT (e.g., SDAP, SARP, SAQP).]
Comment 7: [Language and formatting corrections for grammar, article use, and spacing to improve readability.
]
Response 7: [To enhance readability, the language and grammar in the latest version of the paper have been carefully refined.]

Round 2
Reviewer 1 Report
Comments and Suggestions for Authors
The authors have solved all the concerns.
Author Response
Thank you for your review.
Reviewer 2 Report
Comments and Suggestions for Authors
The authors have addressed some of my previous comments, while others remain unaddressed. The manuscript requires further revision before it can be considered for publication. Please carefully revise the manuscript before resubmission.
More specifically, the attached author response file is simply another copy of the manuscript, whereas a point-by-point response to the review comments is expected. For comment 3, the issue was not a question but information that should be added to the revised manuscript. For comment 5, scale bars are still missing although the authors stated that they have been included.
An additional comment is that abbreviations should be defined when they first appear in the main text, not only in the abstract. It is optional to keep definitions in the abstract, but they are not necessary.
Round 3
Reviewer 2 Report
Comments and Suggestions for Authors
The manuscript has been improved and can now be considered for publication.